# The Role of H-Bonds in the Excited-State Properties of Multichromophoric Systems: Static and Dynamic Aspects

**DOI:** 10.3390/molecules28083553

**Published:** 2023-04-18

**Authors:** Elisa Fresch, Elisabetta Collini

**Affiliations:** Department of Chemical Sciences, University of Padova, Via Marzolo 1, 35131 Padova, Italy

**Keywords:** hydrogen bonds, ultrafast dynamics, 2D electronic spectroscopy, multichromophoric systems, excited-state dynamics

## Abstract

Given their importance, hydrogen bonds (H-bonds) have been the subject of intense investigation since their discovery. Indeed, H-bonds play a fundamental role in determining the structure, the electronic properties, and the dynamics of complex systems, including biologically relevant materials such as DNA and proteins. While H-bonds have been largely investigated for systems in their electronic ground state, fewer studies have focused on how the presence of H-bonds could affect the static and dynamic properties of electronic excited states. This review presents an overview of the more relevant progress in studying the role of H-bond interactions in modulating excited-state features in multichromophoric biomimetic complex systems. The most promising spectroscopic techniques that can be used for investigating the H-bond effects in excited states and for characterizing the ultrafast processes associated with their dynamics are briefly summarized. Then, experimental insights into the modulation of the electronic properties resulting from the presence of H-bond interactions are provided, and the role of the H-bond in tuning the excited-state dynamics and the related photophysical processes is discussed.

## 1. Introduction

Hydrogen bonds (H-bonds) are short-range interactions fundamental for many biological and molecular systems. The importance of H-bonds lies mostly in the fact that they are site-specific interactions with easily predictable orientations, distances, and geometries. Their intrinsic nature suggests that H-bonds can be suitably exploited to tune the static and dynamic properties of electronic states in complex multichromophoric systems. Therefore, the nature of H-bonds and how their presence might affect the structural and electronic properties of these systems have been the subject of intense investigations by different experimental and theoretical methods [1,2,3,4,5,6,7,8].

For many years, the research focused mainly on investigating H-bonds by merely considering their structural function. Indeed, H-bonds provide most of the directional interactions that promote and stabilize protein folding, protein structure formation, and molecular recognition [9,10]. For example, it has been proposed that the formation of a H-bond between the chlorophyll b formyl group and its protein surroundings (as exemplified in Figure 1c) plays a pivotal role in stabilizing different light-harvesting complexes [11]. Moreover, it has been reported that H-bonds also contribute to the protein catalytic mechanisms [12,13] and the ligation of cofactors necessary to accomplish their physiological function [14,15].

From the theoretical point of view, the nature of H-bonds is still the subject of debate, since it can be described as the result of a complex interplay between different energy terms, whose mutual importance depends on the molecular system [16]. Classically, H-bonds can be described as an electrostatic interaction between an electronegative atom and a hydrogen atom attached to a second electronegative atom [17]. However, this interpretation is insufficient to describe many experimental and theoretical observations. Actually, the description of H-bonds requires different components besides the pure electrostatic term, such as charge transfer interactions, π resonance interactions, cooperative effects, etc. [16]. Thus, quantum chemical calculations such as density functional theory (DFT) or post-Hartree–Fock methods are usually employed for a correct description of the H-bond properties [18,19]. Moreover, various energy decomposition schemes are available to identify the different energy components contributing to the H-bond, calculate them with the proper method, and associate them with the right phenomenon [20].

In addition, H-bonds’ networks can be formed around molecules in solvents such as water or alcohols, which can alter the dynamics of molecules or drive protein folding in crowded environments [21]. The stability of the H-bond network is determined by the donor−acceptor free energy landscape, which depends on several factors, such as the chemical nature of the donor and acceptor molecules, orientations, and the delocalized interactions that define the H-bond networks. Thus, molecular dynamics (MD) simulations are often used to obtain a complete molecular description of a solvation system. Indeed, the exact positions of all atoms as a function of time can be extracted, and in turn, molecular information such as the H-bond dynamics at selected positions or the temporal evolution of the H-bond network can be obtained [21].

One of the most important observations emerging from these increasingly sophisticated theoretical descriptions is that H-bonds are not only a mere structural element but also significantly contribute to the definition of the electronic structure, both from the ‘static’ and the ‘dynamic’ points of view. On the one hand, the presence of H-bonding interactions could tune the ‘static’ electronic properties of molecules, such as transition dipole moments, transition energies, and electronic couplings. Indeed, the photoexcitation of an H-bonded system is rapidly followed by a geometric, and hence electronic, reorganization of the H donor and acceptor molecules due to the significant difference in the charge distribution of different electronic states [1]. This process might lead to a weakening or a strengthening of the H-bond in the excited state with respect to the ground state, which in turn has important implications for the modulation of electronic properties. These changes in the ‘static’ electronic properties could also affect the photochemistry and the reactivity of different molecules to the point that H-bonds can be exploited to drive and control specific reaction pathways [22,23,24].

On the other hand, this charge distribution reorganization can also significantly alter the ultrafast excited-state dynamics [25], which has a considerable impact on the efficiency and mechanisms of several photophysical processes, such as photoinduced electron transfer reactions in biological photosynthesis [26,27], artificial photosynthesis [28,29,30], or organic photovoltaics [29,31]. While ground-state dynamics have been the subject of intense study [32,33], only in recent years have the H-bonds dynamics in the excited state started to gain more attention. Generally, these dynamics occur on ultrafast timescales of hundreds of femtoseconds. Therefore, time-resolved spectroscopic techniques with femtosecond resolution must be used to directly monitor the ultrafast dynamic behavior of H-bonds in the excited state [8,34]. Moreover, since these dynamics seem to be predominantly determined by the nuclear motions of the hydrogen donor and acceptor groups [6,35], spectroscopic techniques capable of identifying the frequency and time features of the vibrational modes more strongly coupled with the electronic transition are needed.

This review is focused precisely on the discussion of the experimental methodologies that can be employed to explore the role of H-bonds in the static and dynamic properties of molecular systems and the associated spectroscopic observables.

We first present an overview of some of the most promising spectroscopic methods employed to characterize the static and dynamic aspects of H-bond interactions in excited states (Section 2). The purpose is not to provide a complete list of all the techniques that have been exploited to investigate the multifaceted properties of H-bonded structures but to highlight the ones that can potentially lead to more exhaustive information on H-bonds’ properties in the excited state. Then, we delve into the role of H-bonds in modulating the spectroscopic properties of different molecular complexes (a few examples are sketched in Figure 1) by reviewing several studies on the topic. In particular, we classify the different phenomena as ‘static’ (Section 3) and ‘dynamic’ (Section 4).

One of the most important perspectives emerging from our investigations is that the suitable design of H-bond networks is a particularly powerful tool to drive and control the ultrafast dynamics in complex multichromophoric materials. Therefore, the characterization of excited-state H-bond dynamics is a crucial preliminary step for all those applications requiring the capability of tuning ultrafast photophysical processes.

**Figure 1 molecules-28-03553-f001:**
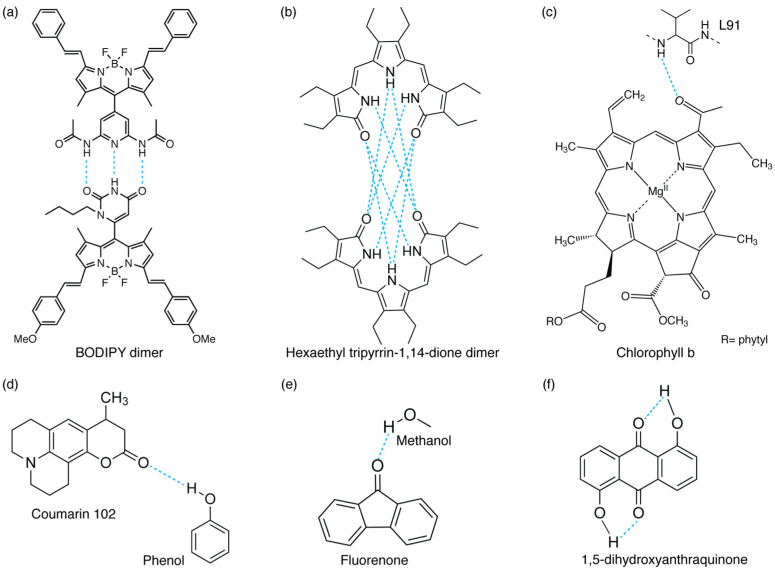
Different examples of inter- and intramolecular H-bond interactions: (**a**,**b**) intermolecular H-bonded dimers (refs. [5] and [8], respectively), (**c**) H-bond between a chlorophyll b chromophore and a protein’s residue (L91) in WSCP protein [4], (**d**,**e**) H-bonds between functional groups and solvent molecules (phenol [36] or methanol [37]), and (**f**) intramolecular H-bonded molecule [6].

## 2. Experimental Techniques

Several spectroscopic techniques have been successfully applied to monitor all the processes directly linked to H-bond interactions. As discussed before, H-bonds’ dynamics are driven by the nuclear motions of the hydrogen donor and acceptor groups [6,35], and therefore, spectroscopic techniques sensitive to vibrational modes are the best reporter of these effects.

### 2.1. Resonance Raman Spectroscopy

Regarding the investigation of ‘static’ properties, one of the most informative techniques is resonance Raman (RR) spectroscopy. RR spectroscopy is typically used to characterize vibrational modes and their coupling with electronic transitions. RR is a third-order technique that requires the interaction of the system under investigation with three electric fields, whose frequency is tuned to be resonant with one of the electronic transitions of the system (Figure 2a). RR thus provides information on the modes more strongly coupled with the electronic transition specifically excited [38]. Since RR spectroscopy is highly sensitive to modifications in the local environment, it allows for correlating any changes detected in the Raman spectra to the atoms or bonds specifically affected by H-bonding interactions. More specifically, it is expected that when a functional group is involved in an H-bond, the corresponding signals in the RR spectrum undergo intensity changes and shifts in energy [38,39]. A typical example of this behavior is provided by the study of the stretching modes of the carbonyl group in biologically relevant molecules, whose major contributions appear in the high-frequency region of the Raman spectra and whose associated frequencies are particularly sensitive to environmental changes, including the formation of H-bonds with surrounding groups [40,41,42].

While these experiments can supply valuable information on the static energetic aspects connected with the establishment of H-bonds, they cannot provide dynamic insights on the intrinsic timescales of the associated charge redistribution in the excited states. The challenge of unraveling excited-state H-bond dynamics is rooted in two main aspects: (i) the complexity of the excited-state potential energy landscape, which is strongly affected by coupled electron and nuclear motions, significantly complicates the characterization from both experimental and theoretical points of view; (ii) in many systems, these dynamics are connected with electronic redistribution and site-specific structural rearrangements and thus occur on a sub-picosecond (sub-ps) to ps timescale.

### 2.2. Time-Resolved Raman Spectroscopy

Time-resolved resonance Raman spectroscopy (TRRS) in different experimental configurations can be employed to explore the kinetics of these processes [43]. TRRS can obtain structural, kinetic, and molecular interaction information by recording RR spectra in a short time span. A widely used TRRS configuration exploits a pump–probe scheme [44,45]. In this experiment, the first pulse (the ‘pump’) promotes an electronic transition, while the second pulse (the ‘probe’) generates the RR scattering effect. A kinetic profile can thus be acquired by measuring the RR spectra for a sequence of different time delays between the two laser pulses. In particular, it is widely used to analyze transient molecular species by directly monitoring changes in the frequency and in the intensity of their vibrational modes to obtain information about their dynamics [46].

In the context of H-bond study, TRRS experiments have so far been successfully applied, for example, to study ion-induced perturbation of water’s H-bonding network [47], the breaking of interhelical H-bonds during the first protein motion following CO dissociation from hemoglobin [48], and H-bond making and breaking dynamics in solute–solvent systems [49]. The main limitation of such experiments is the time resolution, which is intrinsically limited by the time–energy uncertainty principle imposing that for optical pulses, the product of the temporal width and the spectral width cannot be smaller than a specific limiting value, i.e., the Fourier transform limit. For instance, the narrowest possible spectral width for a Gaussian pulse of 1 ps is about 15 cm^−1^. The temporal width of 1 ps is typically indicated as the smallest practical limit for obtaining spontaneous Raman spectra with a reasonable spectral resolution [50].

However, the dynamics of all the processes involving H-bond interactions generally occur on faster timescales, ranging from tens of femtoseconds to a few picoseconds. Therefore, a femtosecond time resolution is needed to directly monitor the ultrafast dynamic behavior of H-bonds in the excited state. To achieve fs resolution, femtosecond stimulated Raman spectroscopy (FSRS) was developed. FSRS exploits stimulated Raman scattering to provide vibrational and structural information with high temporal (sub-50 fs) precision and high spectral (10 cm^−1^) resolution. Traditionally, FSRS is a three-pulse experiment. As in the previous case, a pump pulse excites the molecule, and then the Raman spectrum is read out at various time delays by a sequence of a narrowband Raman pump pulse and a broadband probe pulse. While in TRRS the fundamental resolution limit is dictated by the time bandwidth product of 15 ps·cm^−1^_,_ in FSRS, the experimental parameters that determine the temporal and spectral resolutions are not a conjugate pair of variables. Therefore, the spectral resolution is mainly determined by the bandwidth of the Raman pump pulse, and the effective time resolution is determined by the cross-correlation width of the Raman probe and the pump pulse [51].

FSRS enables the real-time tracking of photophysical and photochemical processes, being able to monitor all the structural changes for Raman active modes while exploiting resonance enhancement in the study of reaction dynamics [52]. Applications of FSRS are varied, ranging from excited-state proton transfer in the green fluorescent protein [53] and excited-state isomerization in cyanobacterial phytochromes [54,55] to the structural dynamics of proteins [56]. Regarding the study of H-bonding interactions, FSRS has been used, for example, to reveal the initial H-bond dynamics following electronic excitation of the coumarin 102 fluorophore (Figure 1d) in ethanol. Direct observation of the H-bond cleavage and its subsequent reformation was captured [49]. In the same work, the authors complemented the FSRS measurements with transient absorption spectroscopy, a technique utilized to measure the photogenerated excited-state absorption energies and associated lifetimes of molecules, materials, and devices [57] and particularly suited also to the characterization of excited-state H-bond dynamics.

### 2.3. Time-Resolved Fluorescence Spectroscopy

The techniques seen so far have made it possible to investigate the effects of H-bonds by monitoring changes in the vibrational modes of molecules. However, the presence of H-bonds can also deeply influence the dynamics of electronic excited states. Thus, the use of techniques sensitive to changes in the excited-state relaxation kinetics could be particularly useful for the characterization of the details regulating the excited-state hydrogen bonding dynamics. In this context, time-resolved fluorescence spectroscopy is a particularly useful technique. It investigates the dynamics of molecules by measuring the decay of the total fluorescence intensity right after photoexcitation. The technique allows for obtaining the fluorescence lifetime, defined as the average time that the molecule remains in the excited state. The fluorescence lifetime of a molecule depends on its local environment, so different events such as solvent reorganization, conformational changes, or quenching that deactivate the excited state also strongly affect the lifetime [58]. It is widely reported in the literature that H-bond interactions have a profound effect on the fluorescence quenching of the excited states of various molecules [59,60,61,62,63,64], and different mechanisms have been proposed to explain this phenomenon. For example, intermolecular H-bonds could enhance the deactivation of the excited state through internal conversion, acting as an efficient accepting mode for radiationless deactivation processes [59,62]. Therefore, the measure of the fluorescent lifetime represents a good reporter of the effects of H-bonds on the excited-state dynamics.

### 2.4. Transient Absorption Spectroscopy

As with time-resolved fluorescence spectroscopy, transient absorption spectroscopy (TAS) also allows for studying the effect of H-bonds on the dynamics of the excited states. Indeed, TAS monitors the dynamics of the optically excited states of a system, probing variations of the differential absorbance over time [65]. In a TAS experiment, a first femtosecond ‘pump’ beam with high optical fluence is used to excite the sample. A second ‘probe’ beam, with a weaker optical fluence, spatially overlaps the pump beam on the sample (Figure 2b). The experiment is based on measuring the variation of the intensity of the probe beam in the absence and presence of the pump pulse as a function of the time delay between the two pulses. This variation is then converted into a differential absorption (Δ*A*), which is a function of the wavelength of the probe and of the time delay between the pump and the probe pulses.

The differential absorption is calculated as ∆*A*(*t*,λ) ∝ − *log*(*I*(*t*, λ) − *I*_0_(λ))/*I*_0_(λ), where *I*_0_(λ) and *I*(*t*, λ) are the intensity of the signal at probe wavelength λ without pump excitation and at a time delay *t* after pump excitation, respectively. The collected data can then be shown in terms of transient absorption (TA) spectra (∆*A* vs. λ) at fixed values of delay time *t*, or in terms of decays (∆*A* vs. *t*) at fixed values of probe wavelengths λ.

Thus, a TA spectrum plots the differential absorption ∆*A*(*t*, λ) as a function of the probe wavelength λ at a fixed value of the time delay *t* after pump excitation. In such a spectrum, negative Δ*A* features can reflect either stimulated emission (SE) from the excited state or ground-state bleaching (GSB) processes. Positive Δ*A* signals are instead the result of excited state absorption (ESA) from an excited state to a higher one.

These spectra and their time evolutions contain information on the dynamic processes contributing to excited-state relaxation dynamics, such as electron, proton, or energy transfer; intersystem crossing; internal conversion; or radiative emission [66]. As shown in Figure 2b, when a molecule is impulsively excited by a broadband fs pulse, vibrational wavepackets of Franck–Condon active modes can also be launched in the ground- and excited-state potential energy surfaces. Therefore, photoexcitation by the pump pulse, in addition to the electronic population dynamics, also generates vibrational coherence dynamics, manifested in the TA spectra as beatings in the signal amplitude. Whether vibrational wavepackets are generated in the ground or excited state depends on the properties of the exciting pulse, and various analysis protocols have been elaborated to make this assignment [67]. It is thus possible to perform a kinetic intensity analysis of vibrational marker bands associated with the dynamic evolution of the groups involved in H-bond formation [49].

### 2.5. Multidimensional Optical Spectroscopies

Besides TAS, multidimensional optical spectroscopies are gaining increasing interest for investigating the dynamics of complex systems. Although they have several similarities with conventional monodimensional spectroscopies such as TAS, their potential lies in the possibility of disentangling signal contributions usually hidden in traditional spectra by separating and resolving the different components of the final signal as a function of frequency in a 2D spectrum. A comprehensive explanation of the theoretical background behind multidimensional spectroscopies and the interpretation of the multidimensional responses can be found in Refs. [68,69,70,71,72]. Briefly, in a 2D experiment, the final signal is measured as a function of the time delays between three exciting pulses. The information encoded by the signal can be more easily visualized in the form of a two-dimensional frequency–frequency map evolving along the population time, where the *x*-axis and *y*-axis correspond to the Fourier transform of the first and third time intervals, respectively, conventionally labeled as excitation and emission frequency. The signal distribution at diagonal and off-diagonal positions in the 2D map provides information on the energy of the levels addressed by the laser excitation and the presence of coupling mechanisms. Indeed, one of the main assets of 2D spectroscopies with respect to analogous 1D techniques such as TAS is that the couplings between different states or transitions are mapped as cross-peaks, far from the diagonal region where the remaining relaxation dynamics occur. Cross-peaks are achievable only in multidimensional techniques. They are the “smoking gun” witnessing the presence of interactions and couplings between states. It is also possible to follow their time behavior and assess the presence of couplings and the associated dynamics. Moreover, the shape of the peaks and the linewidths along the diagonal and anti-diagonal directions depend on the coupling with the environment and are typically exploited to assess inhomogeneous and homogeneous broadening mechanisms, respectively. Finally, information on the dynamic evolution of the system can be retrieved by looking at how the 2D map evolves during the second time interval (‘population’ time). In particular, the technique is sensitive to coherent dynamics manifested as oscillations of the signal amplitude at specific coordinates of the 2D map, as discussed below [68,72,73,74,75].

Two-dimensional infrared (2DIR) and two-dimensional electronic spectroscopy (2DES) are two examples of multidimensional techniques, able to investigate, respectively, the vibrational and electronic properties of a given system (Figure 2c,d) [76]. Two-dimensional infrared is typically used to determine the structure of complex molecular systems by measuring vibrational couplings and to determine their dynamics by looking at structural changes and fluctuations [77,78,79]. Cross-peaks in a 2DIR spectrum are informative of the timescale associated with chemical exchange, structural changes, or anharmonic coupling [80]. Two-dimensional infrared spectroscopy has been fruitfully employed to investigate, for example, the effect of the weakening of an H-bond on the β-turn dynamics [81], the activation of H-bond making and breaking and its temperature dependence [82], and the modulation of vibrational coupling due to the difference in the H-bonds’ strength [83]. However, it must be emphasized that all these processes occur in the electronic ground state (Figure 2c). To extend its use to electronically excited states, it becomes necessary to introduce an additional laser pulse resonant with the molecular transition of interest [84], which leads to greater technical complexity.

Conversely, 2DES provides direct information on the excited states’ dynamics. Diagonal peaks in 2DES maps can be associated with ground-state bleaching (GSB) and stimulated emission (SE) signals, accounting for the time evolution of the population in the ground and excited states, respectively. The relaxation between energy levels or energy transfer between different molecules results in the formation of rising cross-peaks below the diagonal with simultaneous decay of the diagonal signals associated with the initial states. Excitation and emission coordinates of the cross-peaks provide direct information on the energy of the involved states [85].

Moreover, 2DES can detect coherent dynamics, manifested as oscillations of the signal amplitude at specific coordinates of the 2D map. Vibrational or electronic coherences can be distinguished depending on the character of the involved states and the experimental conditions. Indeed, the study of the frequency, amplitude distribution, and dephasing time of such oscillations allows a complete characterization of any coherent dynamics (electronic or vibrational) taking place during the system relaxation [86,87].

For example, the signatures arising from vibrational coherences of ground and excited states occur in distinguishable positions in a 2D map, forming a characteristic ‘chair-like’ pattern of five contributions [88]. More specifically, it is possible to associate beatings at specific coordinates of the 2DES spectrum to a vibrational coherence in the ground or excited state [38]. This also allows a direct comparison with resonant and non-resonant Raman spectra to help identify the vibrational modes more strongly coupled with the electronic transition [38,89].

Two-dimensional electronic spectroscopy has been employed so far to characterize the relaxation and transport dynamics of a wide range of complex multichromophoric systems (from protein light-harvesting antennas and biomimetic molecular artificial complexes to inorganic solid-state materials) [85,90]. We recently proposed exploiting this technique for the investigation of excited-state H-bond dynamics [4,5]. Indeed, the multidimensionality of the technique and its simultaneous frequency and time resolution [70,85] allow for capturing the vibrational motions involved in H-bond dynamics with an unprecedented level of detail, as will be discussed in the following sections.

**Figure 2 molecules-28-03553-f002:**
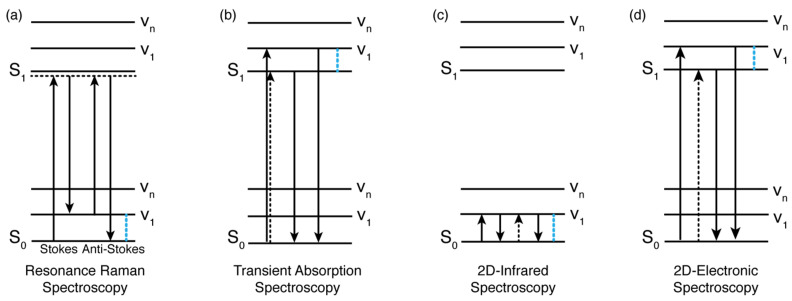
Level diagrams illustrating some of the main processes promoted in the techniques discussed in Section 2. (**a**) Resonance Raman spectroscopy (left, Raman Stokes; right, Raman anti-Stokes). (**b**) Transient absorption spectroscopy (stimulated emission). (**c**) Two-dimensional infrared (2DIR) and (**d**) 2DES (stimulated emission). S_0_ and S_1_ indicate the ground and excited states, respectively, while v_1_-v_n_ identify vibrational levels. Note that the diagrams in (**b**,**d**) differ only for the time sequence of the exciting pulses (in (**b**), the first two interactions are simultaneous and constitute the ‘pump’ pulse, while in (**d**), they are separated in time (see also ref. [85])). As highlighted by the blue dashed lines in (**a**,**c**), only vibrational modes in the electronic ground state can be investigated, while in (**b**,**d**), vibrational modes in the excited states can also be addressed. More details on the interpretation of these diagrams can be found in ref. [91].

## 3. Role of H-Bonds in Modulating Electronic Properties

As previously anticipated, the establishment of H-bond interactions could significantly modify the electronic properties of the excited state upon photoexcitation [92]. Both intra- and intermolecular H-bonding interactions are involved in this phenomenon.

### 3.1. Shift of Electronic Spectral Peaks in Protic Solvents

Zhao et al. demonstrated for the first time that the presence of H-bonding interactions could manifest as a red- or blue-shift of the electronic spectral peaks [93]. To support this claim, they theoretically studied how an intermolecular H-bond affects the photophysics of two thiocarbonyl chromophores, 4H-1-benzopyrane-4-thione (BPT) and thiocoumarin (TC), in methanol (MeOH) [93]. Both these chromophores can form an intermolecular H-bond between the thiocarbonyl group and the O-H group of the solvent molecule. The theoretical calculations showed that the strength of the intermolecular H-bond C=S···H-O is significantly different in the ground and excited states due to adjustments of the structural conformation of the molecules upon photoexcitation. If the H-bond is strengthened, it can lower the excitation energy of the corresponding excited state and can therefore induce a red-shift in the electronic spectrum.

On the other hand, H-bond weakening can increase the excitation energy of the excited state, which is reflected in a spectral blue-shift [93]. In particular, they found that for the TC-MeOH complex, the electronic transition energies of the T_1_, S_1_, and S_2_ states are blue-shifted in the presence of the intermolecular H-bonding interaction with respect to the isolated TC chromophore. It was thus concluded that the intermolecular H-bond in excited systems is weaker than that in the ground state. For the BPT-MeOH complex, they observed instead that the electronic transition energies were blue-shifted for the T_1_ and S_1_ states and red-shifted for the S_2_ state. This led to the conclusion that, in this case, the intermolecular H-bond should be weakened in the T_1_ and S_1_ states and strengthened in the S_2_ state. These first findings by Zhao et al. paved the way for numerous other theoretical and experimental studies to understand how the electronic properties of different chromophores are modified when H-bonding interactions are involved [3,25,37,94,95]. For example, Wang et al. theoretically investigated the low-lying electronic excited states of the benzonitrile (BN) molecule and its derivatives as hydrogen bond acceptors in a hydrogen-donating MeOH solvent [3]. Additionally, the spectral features of fluorenone (FN) molecules in MeOH (Figure 1e) could be explained by a change in the strength of the intermolecular H-bond between FN and solvent molecules upon photoexcitation [37]. By monitoring the spectral shift of the stretching mode of the H-bonded O-H group in different electronic states, it was found that the intermolecular H-bond, which is relatively weak in the ground state, is significantly strengthened in the electronically excited state after the excitation of the system [37].

### 3.2. Shift of Electronic Spectral Peaks in a Protein Environment

A H-bond could also affect the electronic properties of chromophores embedded in a protein environment. It is reported in the literature that the excited-state energy of bacteriochlorophyll (BChl) molecules is strongly affected by the formation of H-bonds. In particular, the C2 acetyl group in the BChl macrocycle can bind to the protein scaffold by establishing a H-bond. This causes the energy of the S_1_ state of BChls in light-harvesting complexes to be shifted by 225 cm^−1^ at most, depending on the strength of the H-bond [96,97]. In 2020, Llansola-Portoles et al. curiously observed that the Q_y_ state of Chl *a* in photosynthetic proteins was also modulated over a wide range of energies [40]. This was a peculiar trait considering that Chl *a*, differently from the bacteriochlorophyll previously discussed, does not possess the acetyl carbonyl group (molecular structure shown in Figure 3a). The only other group that could be involved in H-bond formation is the ketone group at the C13 position, which did not, however, affect the BChls’ energy levels. In their work, they investigated the LHCII antenna by resonance Raman spectroscopy to evaluate the influence of the H-bond on the energy levels of Chl *a.* They discovered that the strength of the H-bond formed by the C13 keto group within the protein binding pocket changed depending on the polarity of the detergent used for the solubilization [40]. Consequently, the position of the Q_y_ transition was red-shifted to a certain extent, which is related to the strength of the formed H-bonds. Moreover, they also observed that the tuning of the electronic properties was significantly larger for Chl *a* bound in photosynthetic proteins than for isolates Chl *a* dissolved in a solvent. This is because the protein scaffold could fine-tune the relative position and orientation of the pigments and the surrounding amino acids to maximize the effect of this H-bond with respect to an isotropic solvent environment.

We have also recently demonstrated that the formation of specific and directional H-bonds could strongly affect the electronic coupling of pigment–protein complexes (Figure 3). Our work employed the 2DES technique to investigate the ultrafast relaxation dynamics of four pigment−protein complexes obtained by reconstituting two water-soluble chlorophyll-binding proteins (WSCPs) with only Chl *a* or Chl *b* [4]. Each WSCP binds four chlorophyll pigments coupled to form two excitonic dimers [98]. Our experiments highlighted that the energy separations among the excitonic states in both Chl *b* samples were considerably higher than the energy gaps estimated through dipole−dipole interactions, suggesting that the transition dipole moment of Chl *b* assumes a higher value in the protein than in solution. These discrepancies were not observed for Chl *a* samples. It is known that Chl *b* could bind to the protein backbone through the formation of a H-bond between its formyl group at the C7 position and specific amino acids, with or without the mediating action of a water molecule. Thus, we concluded that the presence of this specific and directional interaction could promote significant changes in the transition dipole moment and, in turn, also in the electronic coupling and the excitonic energy gaps of pigments [4]. The same argument was also proposed by Pieper et al. to explain the difference in the burn efficiencies observed in a hole-burning experiment in WSCPs reconstituted with Chl *a* or Chl *b* [99]. In this work, the authors claimed that the presence of the H-bond in the case of Chl *b* leads to a more rigid protein environment, causing higher potential barriers between conformational substates [99].

### 3.3. Intra- and Intermolecular H-Bonding Interactions

In the previous sections, we reported examples of intermolecular H-bond interactions, mainly established between suitable functional groups of a solute organic chromophore with solvent molecules (typically an alcohol). However, similar conclusions were also reached in the case of intramolecular H-bonded systems.

For example, it has been found that intramolecular H-bonds in donor–bridge–acceptor complexes provide new electronic coupling pathways for ultrafast charge transfer reactions [100]. Moreover, numerous computational calculations have been dedicated to the investigation of the role of intramolecular H-bonds in the photophysics of many aromatic systems [101,102].

Intramolecular H-bonding interactions are also gaining increasing interest as an activation strategy in catalytic reactions. The establishment of intramolecular H-bonds between different functional groups belonging to the same molecular structure has indeed been exploited to achieve an increase in reactivity, together with the stabilization and stereoselectivity improvement of structures and asymmetric processes [103].

On the other hand, it has been proved that intermolecular H-bonds can significantly facilitate internal conversions, intersystem crossing, and intramolecular charge transfers [3,35].

### 3.4. Excited-State Proton and Hydrogen Transfer

A particularly interesting family of photoreactions driven by the formation of intra- or intermolecular H-bonds includes excited-state proton transfer (ESPT) and excited-state hydrogen transfer (ESHT). These reactions play important roles in photochemistry and biology and have been explored for more than 30 years [104]. The relation between ESPT and EHPT is not always clear, and several processes previously classified as ESPT are now understood as ESHT [105].

It has been proposed that ESHT, i.e., an electron-driven proton transfer process along hydrogen bonds, could play a pivotal role in ultrafast excited-state deactivation [105,106,107]. The mechanism associated with ESHT can be described as an internal conversion from the excited state to the ground state driven by H detachments. A great number of studies have been dedicated to obtaining a deeper understanding of the theoretical and computational details behind ESHT—i.e., for the mechanism of photocatalytic water splitting [108], the photochemistry of the pyrrole–pyridine complex [109], or the deactivation of 2-aminopyridine clusters [110]. More recently, it has been found that ESPT could also happen in systems bearing multiple H-bonds [111,112,113,114]. Understanding the mechanism of a double transfer is quite challenging given the complexity of the system, but it has been proposed that after photoexcitation, the proton could be transferred in two ways. In one case, one proton is first transferred along the H-bond, and only subsequently is the other proton transferred, giving rise to a sequential mechanism. In the other case, the two protons are transferred simultaneously [113,115]. A complete study on the photochemical features of compounds with possible excited-state double proton transfer is reported in Ref. [116].

## 4. Role of H-Bonds in Excited-State Dynamics

The presence of H-bonding interactions could also significantly affect excited-state dynamics. As seen in the above section, upon photoexcitation of the H-bonded systems, reorganization of the hydrogen donor and acceptor molecules occurs due to the significant charge distribution difference between different electronic states [92]. This reorganization has important implications for the modulation of molecular non-equilibrium processes and significantly affects the ultrafast excited-state dynamics [25]. H-bond dynamics are predominantly determined by the vibrational motions of the hydrogen donor and acceptor groups and generally occur on ultrafast timescales of hundreds of femtoseconds. This section presents an overview of the theoretical and experimental studies used to characterize the excited-state H-bond dynamics for several systems.

### 4.1. Early Proofs That H-Bonds Affect the Excited-State Relaxation Dynamics: The Example of Anthraquinones

In 1982, Inoue et al. first reported that the fluorescence lifetimes of excited aminoanthraquinones (AAQs) in ethanol were affected by intermolecular H-bonds with solvent molecules. In AAQ molecules, the excited states have a strong intramolecular charge transfer nature particularly suited to establish strong intermolecular H-bonding interactions. The large electron density on the carbonyl oxygen of AAQ in the excited state strongly promotes an intermolecular hydrogen-bonding interaction with a nonconjugated molecule such as an alcohol. In particular, it was found that the reorganization and reorientation of the hydroxyl groups of alcohol with respect to the carbonyl oxygen of AAQ resulted in two relaxation pathways, of which one leads to a relaxed emissive state and the other to a relaxed nonfluorescent state [117,118].

The physical intuition was that this intermolecular H-bond could drive the radiationless deactivation to the ground state, and that the electronic excited energy could dissipate through the H-bond as vibrational energy. A similar conclusion was reached in 1985 by Flom et al., who studied the dynamics of three disubstituted anthraquinones through picosecond fluorescence spectroscopy. The three analyzed compounds (1,5-dihydroxyanthraquinone (1,5-DHAQ) (Figure 1f), 1,5-diaminoanthraquinone (1,5-DAAQ), and 2,6-diaminoanthraquinone (2,6-DAAQ)) differed in the extent of intra- or intermolecular H-bond contribution involved in the internal conversion mechanism and were chosen to elucidate the specific contribution of intra- and intermolecular H-bonds in driving the excited-state dynamics. It was found that the two molecules with intramolecular H-bonds, 1,5-DHAQ and 1,5-DAAQ, exhibit rapid internal conversion in a broad range of solvents, while 2,6-DAAQ, which cannot form intramolecular H-bonds, exhibits rapid internal conversion in alcohols only [6].

### 4.2. Femtosecond Dynamics of H-Bonds in the Excited State

Elsaesser, Nibbering, and co-workers deeply explored the ultrafast H-bonding dynamics of the coumarin 102 (C102)-phenol complex (Figure 1d) through both femtosecond vibrational spectroscopy and photon echo experiments. In their ultrafast IR experiments, they observed that the vibrational absorption of the C=O band involved in the H-bond underwent a spectral blue-shift within the first 200 fs, ascribed to the H-bond cleavage induced by electronic excitation. To verify the effects of this breakage on the excited-state dynamics, they performed pump–probe experiments in resonance with the electronic transition of C102. They demonstrated that upon photoexcitation, the electronically excited H-bonded state has a finite lifetime and leads to a non-H-bonded complex with a time constant of 170 fs. Thus, since these two pieces of evidence occurred in the same timescale, they concluded that the H-bond could be coupled to the electronic transition and its cleavage could lead to this change in the electronic states [36].

Pines et al. further investigated the role of H-bonding interactions taking place after photoexcitation. Their fs-resolved transient absorption measurements seem to support the previous findings that relaxation processes occurring in the first 100 fs are promoted by a transient change in the H-bonds’ lengths after photoexcitation [34]. Along this line, Palit and co-workers also performed a series of studies on the role of H-bonding interactions in the ultrafast relaxation dynamics of different molecules [119,120,121]. They found that upon photoexcitation, the network of H-bonding interactions between the solute and the solvent molecules undergoes a process of reorganization, which deeply affects the relaxation mechanisms of the excited state, leading to an efficient nonradiative deactivation of the S_1_ state [119,120].

Zvereva et al. theoretically investigated the early-time excited-state photophysics of benzophenone (BP) in water by employing a hybrid quantum mechanics/molecular mechanics (QM/MM) approach. In the first 300 fs, they observed a significant blue-shift of about 4000 cm^−1^ of the ESA band, attributed to the transition from the S_1_ state to higher-energy states. They suggested that this shift is promoted by the stabilization of the S_1_ state due to a water field reorganization around the BP molecule. More specifically, they observed the formation of a very strong H-bond between the BP and one water molecule in a timescale between 100 and 200 fs. This process also seems to be strongly coupled with the vibrational cooling of the excited carbonyl C=O bond, since they observed large amplitudes of the CO vibrations while the water molecule was approaching BP. This evidence was also noticed in the simulated time-resolved 2D electronic spectrum (2DES), where the appearance of line broadening along both the ω_1_ and ω_3_ axes is consistent with the occurrence of vibrational cooling and solvent relaxation at early times [122].

### 4.3. Dynamics of H-Bonded Dimers

Recently, Swain et al. studied the ultrafast electronic dynamics of a tripyrrolic molecule, hexaethyl tripyrrin-1,14-dione (H_3_TD1), in different solvents (Figure 1b). In solvents with sufficient donating or accepting ability, such as pyridine, H_3_TD1 directly forms H-bonds with the solvent and is likely in a monomeric form. However, in solvents with insufficient donating or accepting ability, such as toluene, H_3_TD1 molecules tend to form an intermolecularly H-bonded dimer. The ultrafast dynamics of H3TD1 in both solvents have been investigated by TAS measurements. In the dimeric form, H3TD1 undergoes extremely fast relaxation, likely assisted by the H-bonding interactions that drive the formation of the dimer. In the monomeric form, the molecule undergoes fast internal conversion; however, the relaxation is slower than that observed for the dimer. The shorter timescales may suggest that some of the population proceeds coherently through a conical intersection, with the remaining small amount of the population returning to the ground state more slowly. Thus, it has been proposed that H-bonds can promote rapid internal conversion by acting as radiationless decay accepting modes and can significantly affect the electronic response of conjugated systems from a general perspective, leading to faster S_1_ to S_0_ internal conversion rates [8].

To further check the role of H-bonds in ultrafast relaxation dynamics from a broader perspective, we recently performed 2DES measurements on a H-bonded molecular dimer prepared by the self-assembly of two boron-dipyrromethene (BODIPY) dyes, driven by the formation of a triple H-bond (Figure 1a). We found that the formation of a dimer opens a new ultrafast relaxation channel, characterized by a time constant of about 200 fs, not active in the monomeric BODIPY species. By taking advantage of the multidimensionality of the 2DES technique, two different relaxation processes were identified to contribute to this new kinetic component: firstly, an “intramolecular” relaxation pathway involving the two chromophoric moieties separately, and secondly, an “intermolecular” relaxation mechanism consisting of the transfer of the population from one BODIPY moiety to the other. Even more interesting was the observation that the activation of these new mechanisms is driven by the vibrational motions of the H donor and acceptor groups, which are subject to charge distribution reorganization after photoexcitation. It is worth stressing again here the potential of 2DES among the other spectroscopic techniques since it allows a complete characterization of the vibrational modes more strongly coupled with the excited electronic transitions both in time and in frequency. Different methods can be used to extract information on vibrational dynamics from 2DES data [86,87,123]. The main components contributing to the overall beating behavior of the whole 2D maps can be initially checked by looking at the Fourier spectrum of coherences or the power spectrum, obtained by Fourier-transforming the 2DES maps along the population time t_2_ after integration over the excitation and emission frequencies. It is worth highlighting that only the main components can survive the integration along the two frequency dimensions. In principle, the power spectrum is analogous to a Raman spectrum if only vibrational coherences contribute to the beatings [38,124,125,126]. Figure 4c shows the overall beating behavior of the BODIPY dimer, which, beyond the strong non-resonant contribution of the solvent, is characterized by the presence of three main components (1200 cm^−1^, 1470 cm^−1^, and 1660 cm^−1^) attributed to vibrational modes involving functional groups implicated in the formation of H-bonds in the dimer. Additional information can be obtained by looking at the amplitude distribution of these beating frequencies across the 2D maps (Figure 4a). This can be determined through a global fitting procedure, which indeed allows for analyzing the oscillating components of the signal to retrieve information on the phase, frequency, and amplitude distribution of the beatings (Figure 4b). [123]. The amplitude distribution of a specific beating component can then be visualized in terms of coherent associated spectra (CAS). The CAS associated with the three main vibrational components found for the BODIPY dimer do not show the typical pattern expected for vibrational modes [127] but contribute mainly to the 2DES signal below the diagonal (Figure 4d), where the intermolecular transfer contribution was also contributing. Their dynamic behavior is also peculiar, being characterized by damping times of about 150 fs (Figure 4e), as quantified by the global fitting and visualized through time–frequency transform (TFT) analysis [86]. The advantage of TFT analysis is that it retains information on both the time and frequency dimensions, unravelling the damping dynamics of oscillating signals. In particular, we found that the beating dynamics associated with these modes are damped in the same timescale of the intermolecular transfer process. All these pieces of information point toward an active role of the vibrational motions of the H donor and acceptor groups in the charge redistribution processes following photoexcitation and in the subsequent activation of the intermolecular energy transfer. Note that the activation of an intermolecular mechanism is particularly significant considering the very weak interaction of the two BODIPY molecules and the considerable distance between their centers of mass [5]. This confirms that H-bonds might have a surprisingly strong effect on the energetics and dynamics of chromophores.

## 5. Conclusions

In this review, we present an overview of the main theoretical and experimental investigations on one of the most promising but still underexplored aspects related to H-bond structures—that is, the spectral features and the dynamics of the H-bond in the electronic excited state. On the one hand, we highlighted that the presence of H-bond interactions affects the electronic spectral features of several different systems. First, we clarified the relationship between the electronic spectral shift and the presence of H-bonding interactions in the associated electronically excited state. Indeed, the electronic spectral red or blue shift could be related to a strengthening or weakening of the H-bond upon photoexcitation. The presence of H-bonds also tunes the spectral features of chromophores embedded in a protein environment. The establishment of specific and directional H-bond interactions has very strong consequences for the electronic coupling and energy gaps of pigment–protein complexes, which is quite relevant for modulating their functionalities and dynamics. On the other hand, we reported different examples of how H-bonding interactions could tune the excited-state dynamics. It is well known that several dynamic properties, such as energy relaxation rates, internal conversion, or energy transfer mechanisms, are strongly influenced by intramolecular as well as intermolecular H-bonds.

We would like to emphasize that most works cited here are theoretical investigations. This is partly due to the difficulty of capturing experimentally the fast processes associated with H-bond dynamics. Indeed, excited-state H-bond dynamics usually occur on ultrafast timescales of hundreds of femtoseconds or less, making the characterization of associated mechanisms particularly challenging with conventional time-resolved techniques. In this context, multidimensional ultrafast spectroscopies, particularly 2DES, have proven to be an essential tool for the investigation of these dynamics with both high temporal and spectral resolution. Moreover, it is fundamental to understand the effects of H-bonds on vibrational motions since the excited-state dynamics are usually profoundly influenced by the vibrational modes of the acceptor and donor molecules. Even in this case, the 2DES can be exploited to fully characterize the oscillating behavior through well-established procedures that simultaneously extract information on the frequencies and dephasing times of multiple components.

In conclusion, all the findings reviewed here strongly suggest the possibility of tuning the photophysics and the transport properties of complex materials by establishing specific interactions with the environment. In this scenario, H-bonds are particularly suitable for achieving these modulations because of their directionalities and easily predictable distances and geometries. Additional investigations are certainly needed, but the shreds of evidence discussed here will surely help the future design of H-bonded structures properly engineered for achieving a fine-tuning of the photophysical and dynamic properties of complex functional materials.

## Figures and Tables

**Figure 3 molecules-28-03553-f003:**
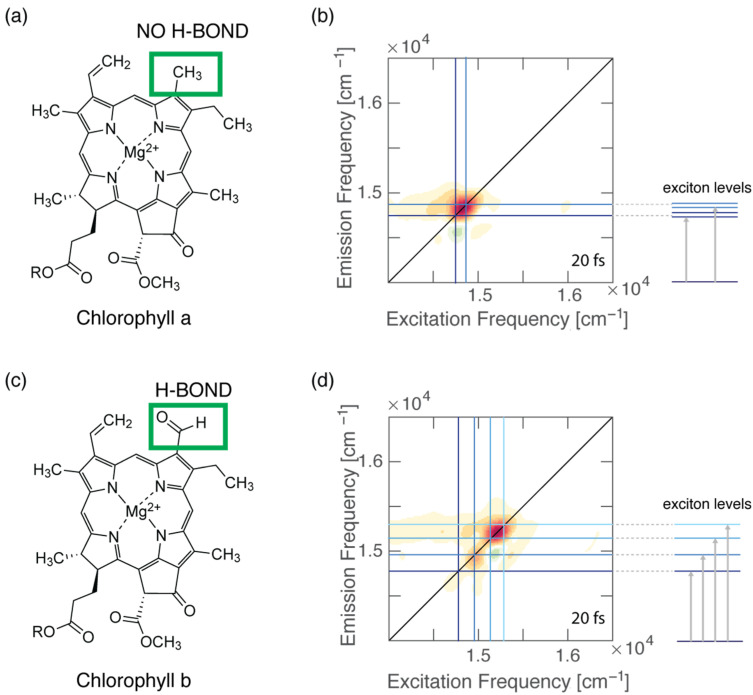
(**a**) Molecular structure of Chl *a*. (**b**) Two-dimensional electronic spectroscopy map of WSCP-Chl a at early time shows that the energy separation between the two excitonic states is in good agreement with the estimation of the coupling based on the dipole−dipole interaction. (**c**) Molecular structure of Chl *b*. (**d**) Two-dimensional electronic spectroscopy map of WSCP-Chl *b* at early time shows a more complex pattern of signals, a signature of excitonic couplings. The energy separations between the exciton levels are greater than those estimated from dipole–dipole calculations.

**Figure 4 molecules-28-03553-f004:**
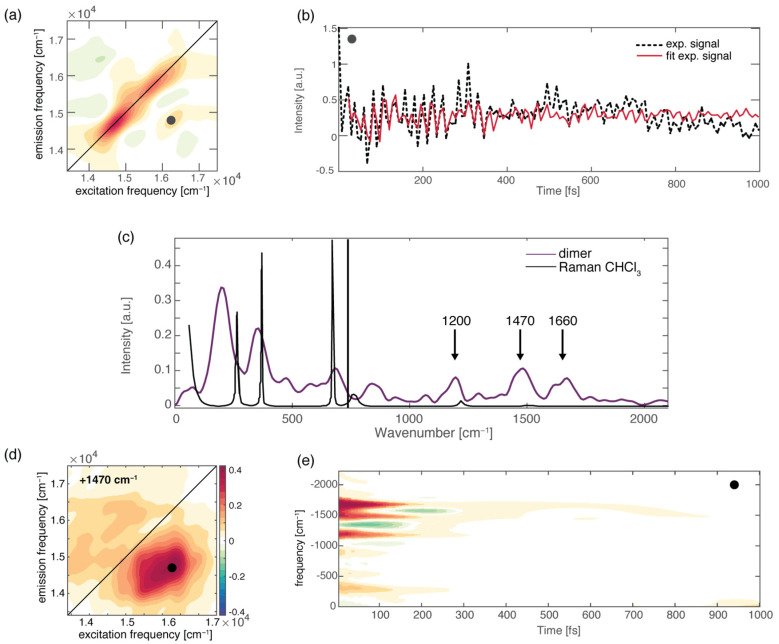
(**a**) Rephasing 2DES map at population time of 100 fs for the BODIPY dimer; the black circle pinpoints the cross-peak signal corresponding to the intermolecular population transfer. (**b**) Signal decay as a function of population time, extracted at the coordinates pinpointed by the circle. Black dashed lines represent the experimental signal, and the solid red lines result from the global fitting procedure. (**c**) Power Fourier spectrum for the BODIPY dimer (purple). The Raman spectrum of the chloroform solvent is also reported for comparison (black line). (**d**) The 2D-CAS plot obtained by the global fitting analysis [123], showing the amplitude distribution of a specific beating component in the 2D maps (+1470 cm^−1^). (**e**) Time–frequency transform of the decay trace shown in panel (**b**), highlighting the frequency (*y*-axis) and the damping times (*x*-axis) of the captured vibrational modes.

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
