# Peer review of "The Role of H-Bonds in the Excited-State Properties of Multichromophoric Systems: Static and Dynamic Aspects"

_molecules, 2023, doi:10.3390/molecules28083553_

Round 1

Reviewer 1 Report

Fresch and Collini claimed to have presented an overview of the main theoretical and experimental investigations on H-bond structures analyzing their spectral features and the dynamics of the H-bond in the electronical excited state. Such a mini-review is potentially interesting but in the end, its potential is not fully expressed. The way the different messages have been arranged is not very helpful for the reader, above all for the one not proper in the field who wants to have an overall picture. 

Indeed, what is really missing, in my opinion, is a theoretical section where the authors could describe the electronic structure properties of the H-bond and how they could influence and be influenced by its environment. Otherwise, the presentation of the experimental techniques loses its sense in this context. 

Moreover, I would find it beneficial for the review the use of specific subsections related to the different classes of cases where the H-Bond is crucial. Otherwise, it results in a series of information that does not help in having a clear and organized picture of the topic. A small introduction to each subsection would help the reader to give the right importance to the reported results. 

Finally, equal importance among all the reported results would improve the balance of the paper.

For the reasons above, the submitted review is not suited to be published in Molecules in the present form. 

Reviewer 2 Report

In this work, the authors tried to give a rather comprehensive review on the role of hydrogen bond in the excited-state processes. The discussions on this topic is a very essential topic in the photochemistry and I think that this work is interesting.

I have some comments here.

1.       The authors mentioned several time-resolved spectra. It may be good to discuss why these techniques are unique. What are the advantages and disadvantages of them. For instance, what different molecular features are relevant to the signals of time resolved RR and TAS? Can we simply use one to replace the second one? Do we obtain different information from them? In which situations, we should use time-resolved RR instead of TAS?

2.       In time-resolved spectra, why do authors not discuss the time-resolved fluorescence spectra? As I know this is also an important time-resolved experimental approaches.

3.       In the discussions of TAS, it may be good to discuss how many different components, such as GSB and so on, can be observed.

4.       In the discussions on time-resolved RR, it may be good to give more emphasize why such approach is unique. Try to give more discussions on the important works of photobiology by Mathies and Fang.

5.       For 2D electronic spectra, it may be good to discuss how many different experimental approaches are available.  Emphasize what different information these groups of spectra can obtain with respect to the normal time-resolved spectra.

6.       Some short and necessary background on the theoretical frameworks on the time-resolved spectra should be given.

7.       The authors mentioned the work by Domcke and coworkers, who also proposed the concept of the excited-state hydrogen transfer (not proton transfer) in many works. It is necessary to discuss this concept and mention more of their works on this topic, including the their electronic-structure and nonadiabatic dynamics calculations, particularly their works on pyrrole-pyridine and water-splitting systems.

8.       More discussions on the intermolecular and intramolecular hydrogen bonds should be given.

9.       Some systems can form the double or multiply hydrogen bonds. Some typical examples on these systems should be given. Discuss whether the proton transfer takes place in the sequential way or at the same time.    

Round 2

Reviewer 1 Report

The authors have improved the readability of the paper and therefore I consider it suitable for publication in Molecules in the present form.

Reviewer 2 Report

The authors properly address my comments and add  the relevant discussions.

Thus I recommend this work to Molecules.